# Effectiveness of a Sustainable Training Program Combining Supervised Outdoor Exercise with Telecoaching on Physical Performance in Elderly People

**Ignazio Leale** [1], **Valerio Giustino** [1,*], **Jessica Brusa** [1], **Matteo Barcellona** [1], **Mario Barbagallo** [2], **Antonio Palma** [1,3], **Giuseppe Messina** [4,5], **Ligia J. Dominguez** [2,6] and **Giuseppe Battaglia** [1,3]

1   Sport and Exercise Sciences Research Unit, Department of Psychology, Educational Science and Human Movement, University of Palermo, 90144 Palermo, Italy; ignazio.leale@unipa.it (I.L.); brusajessica@gmail.com (J.B.); matteo.barcellona@unipa.it (M.B.); antonio.palma@unipa.it (A.P.); giuseppe.battaglia@unipa.it (G.B.)
2   Geriatric Unit, Department of Internal Medicine and Geriatrics, University of Palermo, 90133 Palermo, Italy; mario.barbagallo@unipa.it (M.B.); ligia.dominguez@unipa.it (L.J.D.)
3   Regional Sports School of Italian National Olympic Committee (CONI) Sicilia, 90141 Palermo, Italy
4   Department of Human Sciences and Promotion of the Quality of Life, San Raffaele University, 00144 Rome, Italy; giuseppe.messina@uniroma5.it
5   PLab Research Institute, 90121 Palermo, Italy
6   School of Medicine, University Kore, 94100 Enna, Italy
*   Correspondence: valerio.giustino@unipa.it

**Abstract:** The decrease in functional abilities can negatively influence quality of life and autonomy in elderly people, and physical exercise plays a crucial role regardless of the type. Among the latter, also due to the COVID-19 pandemic, outdoor exercise and telecoaching are settings that have been widely implemented. Hence, the aim of this study was to investigate the effectiveness of a sustainable training program combining supervised outdoor exercise with telecoaching on physical performance in elderly people. A total of 60 participants were recruited and divided into two groups: a trained group (TG) and an untrained group (UG), based on their participation in an 8-week sustainable training program consisted of five sessions/week, which included two sessions/week of supervised outdoor exercise and three sessions/week of telecoaching. Participants were evaluated before and at the end of the training program using the handgrip test, Timed Up and Go (TUG) test, short physical performance battery (SPPB), and Tinetti scale. In the TG, we found a significant improvement in the following tests: right handgrip ($p < 0.001$); left handgrip ($p < 0.001$); TUG ($p < 0.001$); SPPB ($p = 0.01$); and Tinetti scale ($p = 0.006$). A detailed analysis of the SPPB and Tinetti scale showed the lack of significant changes in walking ability: gait speed ($p > 0.05$) and walking parameters in the Tinetti scale ($p > 0.05$). Based on our results, we suggest that a sustainable training program combining supervised outdoor exercise with telecoaching could be effective in the elderly population for improving balance capacity and strength.

**Keywords:** green exercise; outdoor exercise; sustainable exercise; walking program; telecoaching; body balance; postural control; older people; elderly; fall prevention; risk of falls

## 1. Introduction

The aging process is expanding and, in fact, 125 million people worldwide are 80 years old or older [1]. In this regard, the World Health Organization (WHO) predicts an increase in the elderly population which will reach 426 million in 2050 [2]. This process is mainly due to social and economic development, which has caused a reduction in births and an increase in life expectancy [3]. For this reason, the WHO and the European Commission have expressed the need to intervene to improve quality of life and ensure the adequate aging of the elderly population [4,5]. It is well known that aging is a natural and progressive

process which is associated with a decline in both the mental and physical spheres [6]. As a matter of fact, decreased cognitive functions, impaired sensory function, and a reduction in strength, body balance, and flexibility can occur at this stage of life [7–11]. And it is known that, in this population, a minimal decrease in functional abilities can negatively influence quality of life and autonomy [12].

Although healthy aging is a multifactorial phenomenon, regular physical activity and exercise can contribute to preserve the physical and mental components, consequently improving quality of life [13]. The literature suggests that it is necessary to identify effective methods to increase engagement and maintain high training adherence in this population [14,15]. Several studies have demonstrated the fundamental role played by physical activity and exercise in combating heart disease, diabetes, osteoporosis, the risk of falls, and other common deficits that affect the elderly population [16,17]. Battaglia et al. (2020) showed that regular walking in an outdoor environment leads to beneficial effects on body balance capacity in the elderly and this can, consequently, prevent the risk of falls [18]. Similarly, it has been shown that a flexibility training program leads to an improved spinal range of motion (RoM) in a population of elderly women [19]. The effectiveness of training programs on specific abilities and quality of life has been widely demonstrated not only on the healthy elderly population but also on those suffering from pathologies or musculoskeletal alterations [20,21]. A recent study detected improvements in pain and quality of life and significant functional improvement in Timed Up and Go (TUG) performance after a 7-week program in women with multiple vertebral fragility fractures [22]. Hennig et al. (2015) investigated the effects of home-based hand exercises in women with hand osteoarthritis and found positive clinical difference on activity performance measured through the Patient-Specific Functional Scale (PSFS) and significant difference in physical performance such as grip strength [23]. In a study by Martínez-Velilla et al. (2019) a sample of elderly patients was randomly divided into an exercise group (resistance, balance, and walking exercises) and a control group (usual-care hospital care). Among the findings, the exercise group showed significant benefits over the control group, with an increase of 2.2 points detected on the short physical performance battery (SPPB) [24].

This evidence emphasizes the crucial role played by physical exercise regardless of the type. Among the latter, also due to the COVID-19 pandemic, outdoor exercise and telecoaching are settings that have been widely used [25,26]. In detail, outdoor exercise (also named green exercise) concerns the practice of physical activity or exercise in natural environments; it is characterized by easy accessibility and mainly uses walking as a type of exercise [18]. The benefits of nature exposure on psychophysiological health are documented in the literature, and this exercise setting can also counteract conditions caused by urban living such as air pollution, urban noise, and crowding [27,28]. For example, a recent systematic review has shown that Nordic Walking can represent a safe and accessible outdoor exercise for the elderly population, improving cardiovascular function, balance ability, and quality of life. In detail, this walking program showed benefits on dynamic balance, functional balance, upper and lower limb strength, aerobic capacity, and lipid profile [29]. Telecoaching represents a solution for those individuals who are unable to practice physical exercise in sports centers and, thanks to the remote presence of a sports and exercise science professional and the use of wearable devices, it can be individualized [30]. In fact, telecoaching involves the use of information technology and digital tools such as computers and mobile devices to access training services remotely [31]. However, as demonstrated by our previous systematic review, few studies have used telecoaching, as a training method, in the elderly population and no studies have used a combined training approach (telecoaching and supervised exercise) [32]. In detail, Hume et al. (2022) applied this physical exercise method to elderly patients with lung transplantation and, after 12 weeks, the telecoaching group showed statistically significant improvements in daily steps, movement intensity, and quality of life [33]. Von Storch et al. (2019) used telecoaching in elderly patients with type 2 diabetes, showing, after 12 weeks, a reduction in glycated

hemoglobin values, an increase in the number of daily steps, and an improvement in the management of body mass index [34].

Based on this knowledge, the aim of this study was to investigate the effectiveness of a sustainable training program combining supervised outdoor exercise with telecoaching on physical performance in elderly people. This study is part of the project "Walking Leaders", a supervised green exercise program that also includes telecoaching exercises for elderly people. It was developed based on our previous outdoor walking plan in natural environments, named "Passiata Day" model [18].

## 2. Materials and Methods

### 2.1. Study Design

This study is part of the project "Walking Leaders". This is a project developed for elderly people in which a training program combining supervised outdoor exercise with telecoaching is carried out by sports and exercise science professionals. The project was developed based on our previous project, named "Passiata Day" model [18]. The project "Walking Leaders", currently ongoing and lasting a total of 2 years, involves the recruitment of approximately 150 elderly people from the metropolitan city of Palermo (Italy). Over the two years, participants will be recruited in groups of approximately 30 each for a total of 5 groups.

Participants recruited in the study were divided into 2 groups: a trained group (TG), who performed the training program, and an untrained group (UG), who did not complete the established minimum period of the training program (i.e., 75%). As in this study, we did not randomly assign participants to the TG or UG, the present is a quasi-experimental design.

The training program was an 8-week sustainable training program combining supervised outdoor exercise with telecoaching. The training program was preceded by a theoretical phase of 6 meetings, lasting 3 h each, in which the topics of physical activity, balanced diet, adequate rest, and stress management were addressed by medical doctors and professionals of these sectors. For this first study, we considered participants from the first 2 groups recruited.

The measurements were carried out before (T0, baseline) and at the end of the training program (T1, after 8 weeks) by sports and exercise science professionals of the Sport and Exercise Sciences Research Unit of the Department of Psychology, Educational Science and Human Movement of the University of Palermo (Italy), and took place at the Functional Evaluation Laboratory and the Posturology and Biomechanics Laboratory of the same institution.

### 2.2. Participants

A total of 62 participants were enrolled in this study using the following inclusion criteria: (1) ≥60 years of chronological age; (2) ability to provide informed consent for study participation; (3) medical certificate attesting cognitive and physical suitability for the practice of physical activity; and (4) a declared ability of self-care. The exclusion criteria included the following: (1) inability or high difficulty in walking, as established by a score <18 on the Tinetti scale [35]; (2) participating in less than 75% of the training program. Of these, 2 participants dropped out of the study. Of the 60 participants (32 f, 28 m; age: $71.20 \pm 6.01$ years; height: $1.60 \pm 0.09$ m; weight: $70.24 \pm 16.29$ kg; BMI: $27.31 \pm 5.48$), 11 did not complete the established minimum period of the training program (i.e., 75%). For this reason, the 60 participants were divided into 2 groups: a trained group (TG) composed of 49 participants and an untrained group (UG) composed of 11 participants, as shown in Figure 1. The characteristics of the two groups' participants are reported in Table 1.

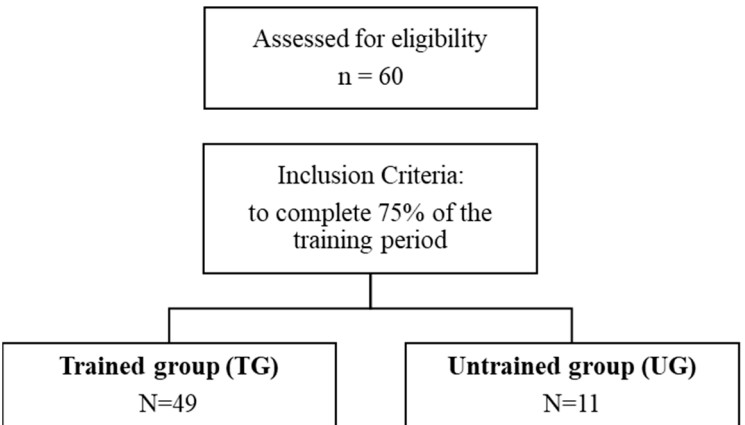

**Figure 1.** Division of participants into the two groups.

**Table 1.** Characteristics of the participants.

|  | Total Sample (Mean ± SD) | TG (n = 49) (Mean ± SD) | UG (n = 11) (Mean ± SD) |
|---|---|---|---|
| Male/Female | 28/32 | 23/26 | 5/6 |
| Age (years) | 71.20 ± 6.01 | 71.32 ± 5.98 | 70.64 ± 6.38 |
| BMI | 27.31 ± 5.48 | 27.22 ± 5.70 | 27.70 ± 4.61 |
| Weight (kg) | 70.24 ± 16.29 | 71.01 ± 14.41 | 72.38 ± 14.23 |
| Height (m) | 1.60 ± 0.09 | 1.60 ± 0.09 | 1.61 ± 0.08 |

Legend. SD, standard deviation.

All participants provided informed consent for participation in the study, and they could withdraw at any stage of the study. The study was approved by the Ethics Committee Palermo 1 of the University Hospital "Policlinico di Palermo" (n. 06/2022) and complies with the criteria defined in the Declaration of Helsinki.

*2.3. Measurements*

All the measurements were carried out in random order for all participants. Each measurement is detailed below.

I.    Anthropometric measurements

Anthropometric measurements were performed as follows. Body weight was measured using a Seca electronic scale (maximum weight recordable: 300 kg; resolution: 100 g; Seca, Hamburg, Germany), with the participants standing upright and barefoot, in light clothing, and with arms along the trunk. Body height was measured through a stadiometer (maximum height recordable: 220 cm; resolution: 1 mm), with the participants standing upright and barefoot. BMI was calculated as weight divided by height squared ($kg/m^2$).

II.    Handgrip test

The handgrip test allows to measure the grip capacity, that is, the maximum isometric strength developed by the muscles of the hand and forearm. Each participant performed the test as recommended by the American Society of Hand Therapists [36], that is, with the back leaning against the back of the chair and the elbow joint positioned at 90°. At the signal, each participant performed the test by squeezing the handle of a mechanical dynamometer (Kern Map model 80K1-Kern®, Kern & Sohn GmbH, Balingen, Germany) with maximum isometric grip strength. Each participant performed 3 trials with both the right and left hand, with 1 min of rest between trials. For the statistical analysis, the best results of the 3 trials for each hand were considered.

III.    Timed Up and Go (TUG) test

The TUG is a test used to measure balance capacity, the level of functional mobility, and the associated risk of falls. The test measures the time it takes to get up from a chair without using the hands, walk 3 m, turn around, return, and sit back down on the chair. During the test, the participant can use walking aids. A score of 10 s or less indicates normal mobility; times between 11 and 20 s represent the normal range for elderly individuals with frailty or disabilities; times over 20 s indicate that the individual requires assistance; values above 30 s indicate a high risk of falls [37].

IV.    Short physical performance battery (SPPB)

The SPPB is a test developed to measure functional status and physical performance [38]. This battery contains 3 components with the aim of analyzing balance capacity, gait speed, and lower limb function.

In detail, for the balance capacity, the following 3 balance tests are included: (1) the "side-by-side stand", in which each participant was asked to stand with feet together, side-by-side, for 10 s; (2) the "semi-tandem stand", in which each participant was asked to stand with the side of the heel of one foot placed by the big toe of the other foot for 10 s; and (3) the "full-tandem stand", in which each participant was asked to stand with the heel of one foot placed in front of the toes of the other foot for 10 s.

As for gait speed, each participant was asked to walk 4 m at a normal pace. The starting time was recorded when one foot began to move beyond the starting line and stopped when one foot completely passed the finishing line.

Regarding lower limb function, to execute the chair stand test, each participant was asked to sit on a chair with both feet flat on the floor, and then, with arms folded across the chest, to stand up and sit down as quickly as possible 5 times without stopping. The execution time was recorded. The test was considered valid if the participant performed the full movement (knees extended) and was stopped if the participant became too tired, used the arms, or the execution exceeded 1 min.

For each component, the maximum total score is 4 and the final maximum score that can be obtained by their sum is 12. A high score indicates a high level of functionality, while a low score indicates a low level of functionality. Low scores on the SPPB have been shown to predict increased fall risk [39], reduced personal autonomy [40], decreased mobility [41], and worsened health [42].

V.    Tinetti Scale

The Tinetti scale allows balance capacity and gait to be measured [43]. The items concerning balance are the following: sitting balance, rising from a chair, attempts to stand, standing balance, standing balance prolonged, Romberg, Romberg with nudge, turning balance (360°), and sitting down. The following items focus on gait: initiation of gait, step length and height, step symmetry, step continuity, path deviation, trunk stability, and walking stance.

The scores of all items are combined to develop three measures: an overall balance assessment score, an overall gait assessment score, and a total score. The maximum score for the balance component is 16 points. The maximum score for the gait component is 12 points. The maximum total score is 28 points. A total score of 24 or more indicates a low risk of falling, a total score of 19 to 24 indicates a moderate risk of falling, and a total score of 18 or less indicates a high risk of falling [35,44].

### 2.4. Training Program

The sustainable training program was a combined approach consisting of supervised outdoor exercise and telecoaching. The sustainable training program lasted 8 weeks and consisted of 5 sessions/week, of which 2 sessions/week involved supervised outdoor exercise (walking, as in the study by Battaglia et al. (2020) [18], and balance circuit ~90 min, as in the studies by Battaglia et al. (2010) and Bellafiore et al. (2011) [45,46]) and 3 sessions/week involved telecoaching [2 sessions/week of resistance training (~40 min) and 1 session/week of free-walking (~40 min)].

In accordance with the WHO recommendations for physical activity practice in elderly people (i.e., from 150 to 300 min of moderate physical activity) [47], participants performed a weekly volume of 300 min of physical activity. More details on the weekly distribution of physical activity minutes are shown in Figure 2.

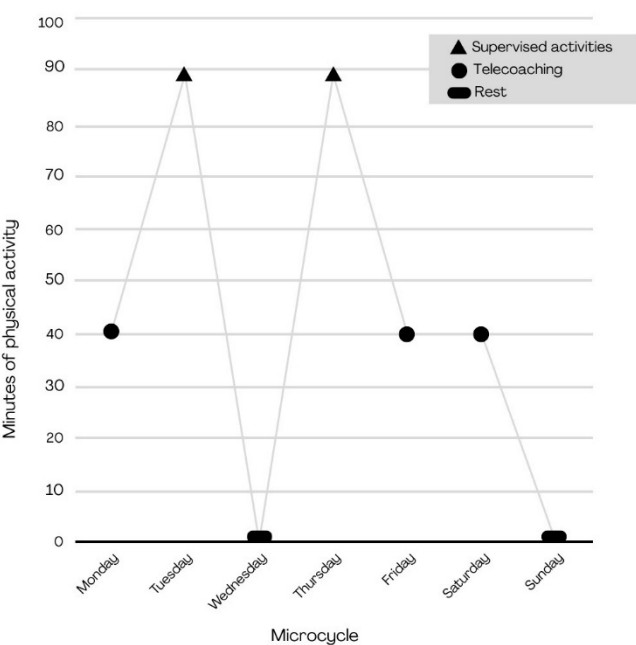

**Figure 2.** Microcycle training program.

I.     Supervised outdoor exercise

Each session of the supervised outdoor exercise was composed of a warm-up phase (~15 min), a central phase (~60 min), and a cool-down phase (~15 min).

In the warm-up phase, from standing posture, participants executed a standard sequence of exercises, including circling movements of the shoulders, pelvis, feet, and trunk rotations.

In the central phase of each session, participants were randomly divided into two subgroups, and one subgroup started with the execution of a walking path and subsequently carried out a balance circuit while the other subgroup performed the exercises in reverse order.

The supervised walking, carried out in groups, involved the following setting: (1) 1st and 2nd week: slow walk (1.5–3.0 km/h; <3 MET); (2) 3rd week: moderate walk (4.5–6.0 km/h 3–6 MET); (3) 4th and 5th week: slow walk (1.5–3.0 km/h) with axial loading (2% of body weight); (4) 6th and 7th week: moderate walk (4.5–6.0 km/h) with axial loading (2% of body weight); (5) 8th week: moderate walk (4.5–6.0 km/h 3–6 MET). The walking speed was controlled by a sports and exercise science professional (J.B.) who was the leader of the group and by another researcher at the end of the group (I.L.) who wore a smartwatch, as well as by at least 3 other sports and exercise science professionals who were in the group. A representation of the supervised walking is shown in Figure 3.

The supervised balance circuit was managed by a sports and exercise science professional (M.B.) and predicted a ratio of 3:8 (3 coaches and 8 subjects) in the initial phase of the program, which evolved with a ratio of 2:8 in the final four weeks. The balance circuit involved the following setting: (1) from 1st to 4th week = 2 circuits training/session; (2) from 5th to 8th week = 3 circuits training/session. For the balance circuit the axial loading adopted from week 4 to week 7 was used.

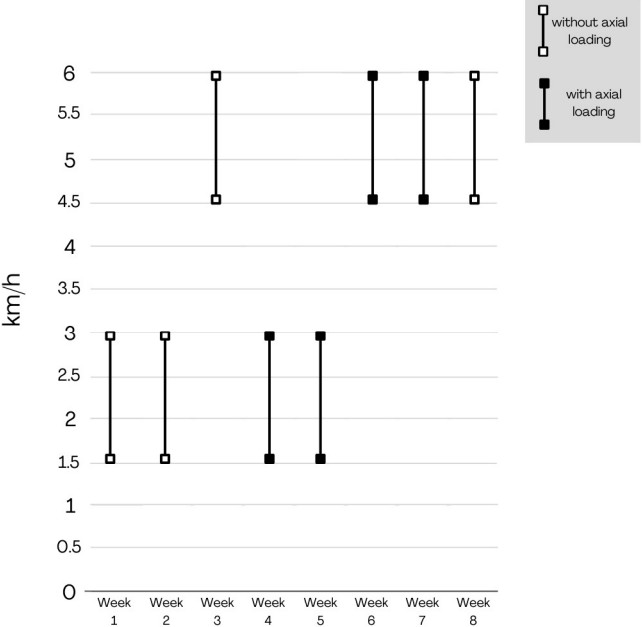

**Figure 3.** Supervised outdoor walking program. X: axial loading.

The cool-down phase included free body movements, deep breaths, and isometric contractions of both arms in different positions.

II. Telecoaching

The telecoaching training, carried out using demonstration videos, consisted of resistance training and free-walking.

The resistance training (2 sessions/week of about 40 min) consisted of a warm-up phase, a central phase, and a cool-down phase.

The warm-up phase included dynamic movements of the neck, shoulders, trunk, and legs (3 series x 8 repetitions).

The central phase included 8 exercises (3 series x 8 repetitions): (a) lateral raises with elastic bands; (b) biceps curl with elastic bands; (c) handgrip exercise; (d) front raises with elastic bands; (e) box squat; and (f) calf exercise.

The cool-down phase included free body movement, deep breaths, and isometric contractions (2 series of 30 s).

The free-walking (1 session/week of about 40 min) was self-managed based on the indications previously received in telecoaching.

Figure 4 shows both the resistance training and the free-walking setting of the telecoaching training.

The telecoaching training was monitored via weekly calls and through an exercise diary filled out weekly by each participant. Moreover, in order to guarantee the correct execution of the training program, a training manual was created. In addition, sports and exercise science professionals were always available to resolve any doubts regarding the execution of the exercises. The presence of psychologists was useful in maintaining high motivation and adherence to the telecoaching training.

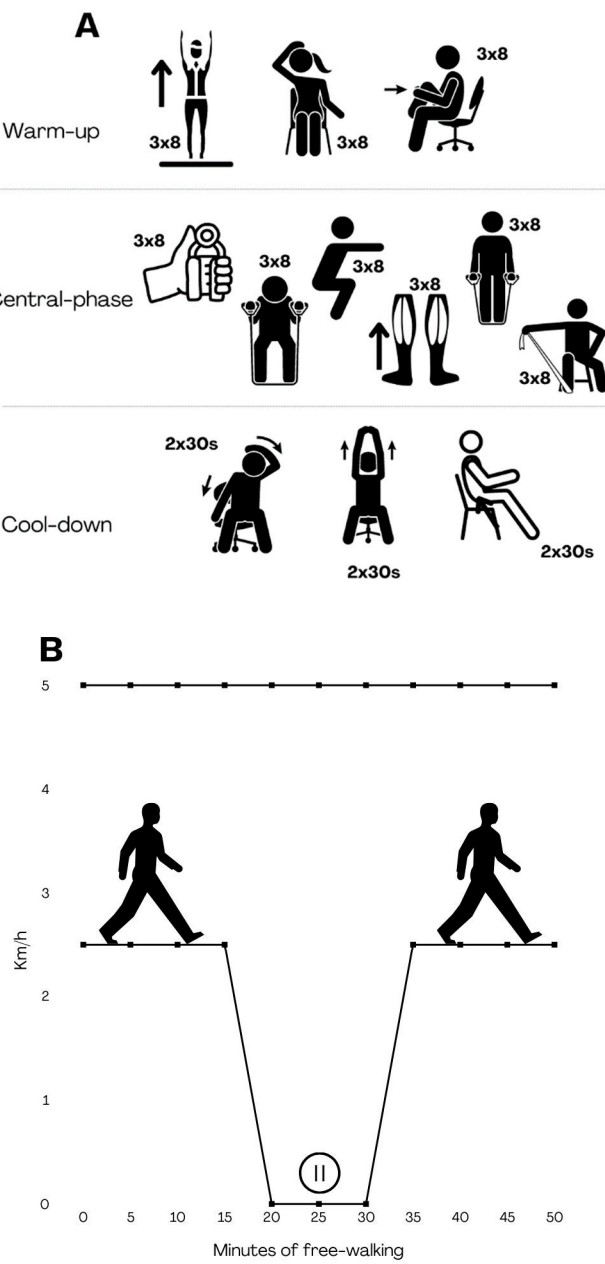

**Figure 4.** (**A**). Telecoaching: resistance training exercises with series and repetitions. (**B**). Telecoaching: free walking.

## 2.5. Statistical Analysis

Data are presented as mean ± standard deviation. The data distributions were tested through the Shapiro–Wilk test. Any difference between T0 and T1 within groups was analyzed. Student's T-test or Wilcoxon T-test was used in relation to the results of data distributions of the Shapiro–Wilk test.

All analyses were performed using Jamovi software (The Jamovi project (2021). Jamovi (Version 1.8.0.1) [Computer Software]). The significance for all analyses was set at $p < 0.05$. Graphs were created using GraphPad Prism 7 (GraphPad Software Inc., San Diego, CA, USA).

## 3. Results

Of the total of 60 participants, 49 performed 75% of the training protocol, resulting in the TG (26 f, 23 m). More information about the TG and UG can be found in Table 1.

The results of the Shapiro–Wilk test of each measurement, means and standard deviations, and levels of significance of the comparison between T0 and T1 are reported in Table 2.

**Table 2.** Performances at T0 and T1 of the TG.

| | Shapiro–Wilk | | T0 | T1 | |
| | W | *p*-Value | (Mean ± SD) | (Mean ± SD) | *p*-Value |
|---|---|---|---|---|---|
| Right handgrip ° | 0.94 | 0.015 | 27.47 ± 11.58 | 28.82 ± 11.06 | <0.001 *** |
| Left handgrip × | 0.967 | 0.176 | 25.11 ± 10.80 | 26.39 ± 10.30 | <0.001 *** |
| TUG ° | 0.918 | 0.002 | 7.19 ± 1.37 | 6.56 ± 1.36 | <0.001 *** |
| SPPB ° | 0.759 | <0.001 | 11.22 ± 0.98 | 11.5 ± 0.82 | 0.012 * |
| Gait Speed ° | 0.838 | <0.001 | 4.90 ± 1.66 | 4.67 ± 1.43 | 0.194 |
| Sit To Stand ° | 0.946 | 0.025 | 9.98 ± 2.07 | 8.05 ± 2.02 | <0.001 *** |
| Tinetti Scale ° | 0.871 | <0.001 | 24.92 ± 2.74 | 26.20 ± 1.38 | 0.002 ** |
| Balance ° | 0.741 | <0.001 | 15.78 ± 1.53 | 16.76 ± 0.48 | <0.001 *** |
| Gait ° | 0.948 | 0.03 | 9.14 ± 1.63 | 9.45 ± 1.26 | 0.359 |

Legend. SD, standard deviation; °, non-normal distribution (Wilcoxon T-test was used for T0 vs. T1 comparison); ×, normal distribution (Student's *t*-test was used for T0 vs. T1 comparison); *, $p < 0.05$ between T0 and T1; **, $p < 0.01$ between T0 and T1; ***, $p < 0.001$ between T0 and T1; TUG, Timed Up and Go; SPPB, short physical performance battery.

Figure 5 shows the comparisons between T0 and T1 for right handgrip (Figure 5A), left handgrip (Figure 5B), TUG (Figure 5C), Sit To Stand of the SPPB (Figure 5D), and the balance of the Tinetti scale (Figure 5E).

In the TG, we found significant improvements in the following tests: right handgrip ($p < 0.001$); left handgrip ($p < 0.001$); TUG ($p < 0.001$); SPPB ($p = 0.012$); and the Tinetti scale ($p = 0.002$). A detailed analysis showed the lack of significant changes in gait speed ($p > 0.05$) and gait ($p > 0.05$) for the SPPB and the Tinetti scale, respectively. No statistically significant changes were shown in the UG ($p > 0.05$).

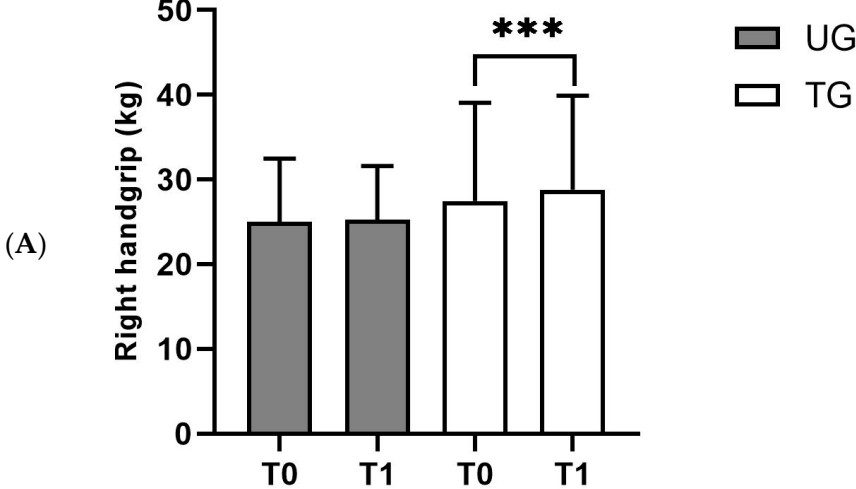

**Figure 5.** *Cont*.

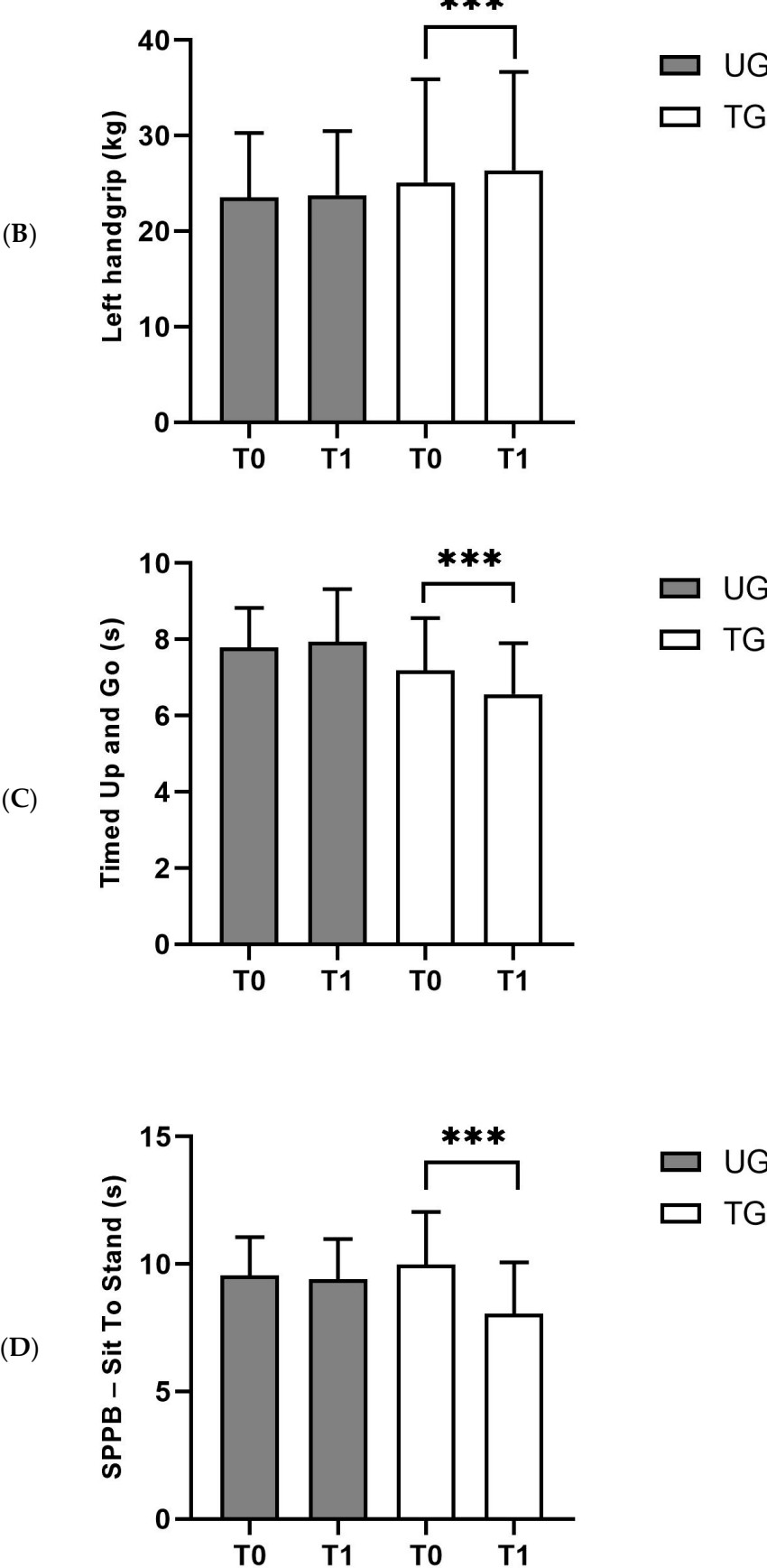

**Figure 5.** *Cont.*

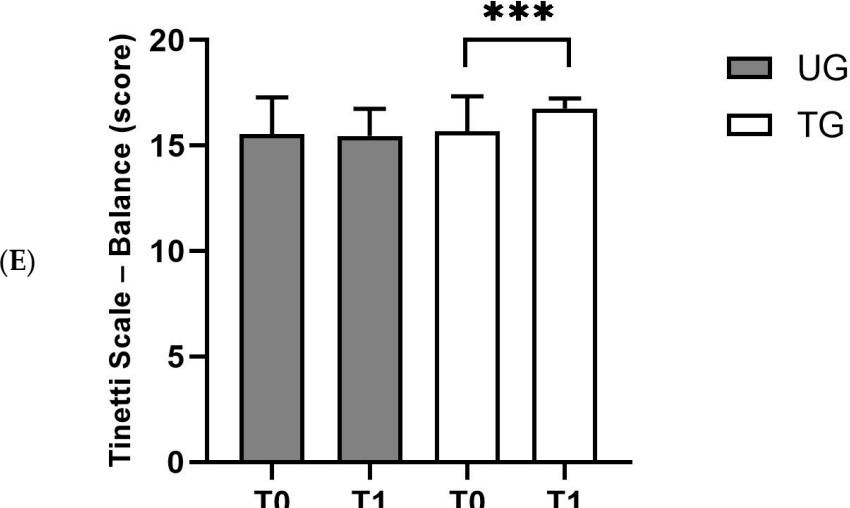

**Figure 5.** Results of the comparisons between T0 and T1: (**A**) kg expressed by the right hand in the handgrip test; (**B**) kg expressed by the left hand in the handgrip test; (**C**) time to perform the TUG test; (**D**) seconds for the execution of the Sit To Stand of the SPPB; and (**E**) score of the balance of the Tinetti scale. ***, $p < 0.001$.

## 4. Discussion

Few studied have evaluated the effectiveness of telecoaching in elderly people [32], and no study has evaluated the effectiveness of a sustainable training program combining supervised outdoor exercise with telecoaching on physical performance in elderly people.

The effectiveness of physical activity and exercise in the elderly population has been demonstrated in numerous studies [48–50]. In fact, beneficial effects have been found on the capacity to balance [45,51] and on the subsequent risk of falls [52], as well as on the contrast of pathologies such as osteoporosis [53,54] and sarcopenia [55,56]. However, as reported by Furtado et al. (2023), it is necessary to implement new methodologies and intervention strategies to increase the involvement of the elderly population and increase adherence to physical activity, minimizing barriers to participation [14]. Indeed, different types of exercise settings exist and among these, also due to the COVID-19 pandemic, outdoor exercise and telecoaching are increasingly implemented.

Our previous study showed that a no-structured outdoor walking plan carried out in natural environments for six-month once a week improved balance capacity in elderly people [18]. These preliminary findings from the "Passiata Day" model led us to develop a further walking program associated with telecoaching named "Walking Leaders".

The results of the present study showed that an 8-week sustainable training program consisting of a supervised outdoor exercise and telecoaching led to significant positive effects on all the outcomes considered. In fact, the TG significantly improved the performances of handgrip, TUG, SPPB, and the Tinetti scale, while no differences were detected in the UG. However, a more detailed analysis highlighted the lack of improvement in walking ability. Indeed, no significant change was found in the component of gait speed and the field gait of the Tinetti scale. This result could depend on the initial choice to exclude individuals with high walking difficulty, which means that, a priori, a sample with a good walking level was recruited. This choice was made because the protocol included the walking exercise. This improvement, as already highlighted in our preliminary study [18], could be induced by carrying out the walking program outdoors, i.e., on non-uniform ground which stimulates proprioception to a greater extent than walking on a regular surface, increasing, therefore, the balance capacity. A fundamental aspect of the walking plan of the present study was the variability of walking speed and axial loading during the program. Indeed, walking speed in elderly people is a topic of interest for researchers, and our results are in line with the existing literature [57]. In this way, in a recent systematic review by

Bai et al. (2022), the effect of brisk walking on various outcomes has been studied [58]. The authors demonstrated that brisk walking improves cardiorespiratory fitness, muscular strength, and body composition. A research by Fan et al. (2016) investigated the influence of gait speed (slow, normal, and fast speed of walking) on walking stability in the elderly, showing that stride length and cadence increased at fast speed [59]. An interesting research by Chatutain et al. (2019) emphasizes the importance of walking among elderly [60]. The authors highlight the practice of walking meditation as a mindfulness practice of walking, that is, focusing on the movements of the legs while walking slowly [60]. In line with our results, a recent pilot study involving people with multiple sclerosis demonstrated that axial loading during walking can influence stride length, cadence, and gait speed [61]. Our results are consistent with these studies for supporting the effectiveness of axial loading during walking plan. It should be noted that the scientific literature shows conflicting results regarding the benefits and risks of walking [62,63]. Indeed, Okupo et al. (2016) showed that, in the elderly without risk of falling, walking improves balance and is therefore a useful strategy for preventing falls [64]. In contrast, Gillespie et al. (2003) found that walking should not be recommended to the elderly population with great difficulties in walking and therefore at high risk of falling [65]. Based on this previous research, it would appear that walking may have negative effects on populations at risk of falling. In accordance with these studies, we checked this aspect by recruiting older adults who had no history of falls and who were not at risk of falling.

Another important result of our study is related to the resistance training carried out in telecoaching. Many studies have shown the effectiveness of resistance training in terms of strength improvement and to counteract some pathologies in this population [66,67]. In a recent study, Otsuka et al. (2022) reported that a resistance training program of 24 weeks improved muscle quantity and quality in older people [68]. Similar results were found by the study of Seo et al. (2021), in which after 16 weeks of resistance training, older adult women with sarcopenia improved functional fitness, grip strength, gait speed, and isometric muscle strength [69]. Among the findings by Hennig et al. (2015) on the effects of home-based hand exercises in women with hand osteoarthritis, there was a significant difference in grip strength [23], in line with our results. The evidence for the effectiveness of telecoaching training is growing. For example, in a sample of breast cancer survivors, telecoaching improved adherence and retention among participants [70]. A study by Demeyer et al. (2017) showed that a 12-week semiautomated telecoaching program, including a step counter and a smartphone app, increased the amount and the intensity of physical activity in patients with COPD [30]. In our study, the improvements in the handgrip test and the absence of adverse events highlight the effectiveness and the safety and feasibility of a resistance training program carried out in telecoaching. Thus, considering that telecoaching is a safe and accessible training modality used in other populations [71–74], this can represent an innovative setting that can also be administered to the elderly in order to achieve the WHO recommendations on physical activity practice for this population.

### 4.1. Strengths and Limitations

The main strength of our study is the adherence to the project. In fact, 81.66% of the participants completed the training program. Other strengths are represented by the fact that this exercise setting is easily accessible, economical, and free of adverse events. In fact, no accidents occurred during the entire training program. Among the strengths, it should be noted that the training program was developed and supervised by sports and sports science professionals, with the collaboration of geriatric doctors and psychologists. Indeed, as demonstrated by Furtado and colleagues, a comprehensive and targeted approach is needed to promote overall well-being in older individuals [75]. One of the main limitations is related to the small sample size of this study. However, this is a first study of the "Walking Leaders" project which, as described above, is ongoing and has a duration of 2 years, and intends to recruit a total sample of approximately 150 elderly people from

the metropolitan city of Palermo (Italy). Among the limitations of the study, we should note the failure to include, by choice, individuals with a high risk of falls, which does not allow for the generalization of the results. Another limitation is represented by the absence of a follow-up that aims to highlight any change in the practice of physical activity and exercise of the elderly who participated in the project. The absence of a random assignment of participants to the two groups, a process that makes our study quasi-experimental, may represent a further limitation of the study.

### 4.2. Practical Implications

This study could have a broad practical application for the elderly population who wants to practice physical activity and exercise, given the easy accessibility to outdoor physical exercise, even in urban areas, and its economic convenience. Likewise, the convenience of practicing exercise from one's own home, such as using telecoaching, could help to achieve the physical activity levels recommended by the WHO. These physical exercise settings could also have an interesting implication for sports and exercise science professionals. In fact, by using both green exercise and telecoaching, they could more easily reach the elderly population who wants to practice physical exercise. Moreover, these exercise settings allow greater flexibility, and do not imply the use of a sports building or specific tools, emphasizing their cost-effectiveness. Furthermore, as regards the outdoor exercise, the possibility of grouping elderly people with similar characteristics (after measuring their physical and functional abilities) would favor adherence to the training program and the possibility of creating new social relationships for those who participate.

### 5. Conclusions

In conclusion, based on our results, we suggest that a sustainable training program combining supervised outdoor exercise and telecoaching could be effective in the elderly population for improving balance capacity and strength. In addition, this combined method could be an effective intervention strategy to increase the levels of physical activity in the elderly population, maintaining high adherence to training and being safe, effective, and risk-free. Further studies should also evaluate the effectiveness of this type of training on body composition, body posture, and psychological components. Furthermore, we recommend that walking programs are carried out by a sports and exercise science professional and in groups, since physical activity carried out in groups has been shown to be associated with high levels of enjoyment, improving adherence to the training program [76].

**Author Contributions:** Conceptualization, G.M., L.J.D. and G.B.; Methodology, L.J.D. and G.B.; Software, J.B. and I.L.; Validation, V.G., M.B. (Mario Barbagallo), A.P. and G.B.; Formal analysis, I.L.; Investigation, I.L., J.B. and M.B. (Matteo Barcellona); Resources, M.B. (Mario Barbagallo); Data curation, I.L., V.G. and J.B.; Writing—original draft, I.L.; Writing—review & editing, V.G., I.L. and G.B.; Visualization, V.G., J.B., M.B. (Mario Barbagallo) and A.P.; Supervision, A.P., G.M., L.J.D. and G.B.; Project administration, G.M., L.J.D. and G.B.; Funding acquisition, M.B. (Mario Barbagallo). All authors have read and agreed to the published version of the manuscript.

**Funding:** This research was partially funded by the project PSN 2017 action 4.2.1 "Prevenzione incidenti domestici e promozione attività fisica nell'anziano".

**Institutional Review Board Statement:** The study was conducted in accordance with the Declaration of Helsinki, and approved by the Ethics Committee Palermo 1 of the University Hospital "Policlinico di Palermo" (n. 06/2022).

**Informed Consent Statement:** Informed consent was obtained from all subjects involved in the study.

**Data Availability Statement:** Data is contained within the article.

**Conflicts of Interest:** The authors declare no conflict of interest.

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
