# Peer review of "Effectiveness of a Sustainable Training Program Combining Supervised Outdoor Exercise with Telecoaching on Physical Performance in Elderly People"

_sustainability, doi:10.3390/su16083254_

Round 1

Reviewer 1 Report

Comments and Suggestions for Authors

Dear authors, congratulations, this is a very interesting study.
However i think the  manuscript should be improved , specially on the introduction and discussion sections.

The introduction should be increased with more recent studies.

The discussion section should be improved  to reflect all the findings of the study.
Please add more recent references.

Please avoid this sentence "To the best of our knowledge" and reformulate the discussions.

Strengths and Limitations and Practical implications should apear before the conclusions

The conclusions should be increased.

The goal of the study was to investigate the effectiveness of
a sustainable training program combining supervised outdoor exercise with telecoaching on physical performance in elderly people. I consider the topic original and addressed gap in the literature. I consider the methodology well addressed, and no need to improve. The tables and figures are well and balanced

Please see  the following references:

Furtado GE, Reis ASLDS, Braga-Pereira R, Caldo-Silva A, Teques P, Sampaio AR, Santos CAFD, Bachi ALL, Campos F, Borges GF, Brito-Costa S. Impact of Exercise Interventions on Sustained Brain Health Outcomes in Frail Older Individuals: A Comprehensive Review of Systematic Reviews. Healthcare (Basel). 2023 Dec 13;11(24):3160. doi: 10.3390/healthcare11243160. PMID: 38132050; PMCID: PMC10742503.

Furtado, G.E.; Letieri, R.V.; Carballeira, E. Exercise Evaluation and Prescription in Older Adults. Healthcare 2023, 11, 42. https://doi.org/10.3390/healthcare11010042

Author Response

#Reviewer 1

  • Dear authors, congratulations, this is a very interesting study. However, I think the manuscript should be improved, especially on the introduction and discussion sections. The introduction should be increased with more recent studies. The discussion section should be improved to reflect all the findings of the study. Please add more recent references.

Thank you for this comment. According with the suggestions, we revised the relative sections and increased them with more recent studies.

  • Please avoid this sentence "To the best of our knowledge" and reformulate the discussions.

Thank you for this comment. The above sentence was deleted, and the discussion section revised.

  • Strengths and Limitations and Practical implications should appear before the conclusions.

We apologize for this inconvenience. We have modified the order of these sections.

  • The conclusions should be increased.

Thank you for this suggestion. The entire section has been revised and improved.

  • Please see the following references:

Furtado ge, reis aslds, braga-pereira r, caldo-silva a, teques p, sampaio ar, santos cafd, bachi all, campos f, borges gf, brito-costa s. impact of exercise interventions on sustained brain health outcomes in frail older individuals: a comprehensive review of systematic reviews. healthcare (basel). 2023 dec 13;11(24):3160. doi: 10.3390/healthcare11243160. pmid: 38132050; pmcid: pmc10742503.

Furtado, g.e.; letieri, r.v.; carballeira, e. exercise evaluation and prescription in older adults. healthcare 2023, 11, 42. https://doi.org/10.3390/healthcare11010042

Thank you for this suggestion. These interesting studies have been extremely helpful to improve our manuscript and we added them into the references.

Reviewer 2 Report

Comments and Suggestions for Authors

This study is aimed at addressing the important issue of improving the health of the elderly as a guarantee of a long healthy life. This study is relevant and up-to-date.

The authors propose a combination of physical exercises and modern telecommunication technologies during classes. Thus, combining classes in a group with a trainer and independent classes at home, which significantly expands the arsenal of influence.

The authors are guided by a wide range of scientific research in recent years. They use modern research methods to assess the health benefits of the proposed training programme.

The structure of the pedagogical experiment developed by the authors meets the requirements for testing this hypothesis.

The conclusions reflect the research findings.

In addition to the positive aspects of the presented article, there are provisions that require additional discussion:

- In the introduction, outline the range of unresolved issues that underpin the purpose and objectives of the article.

- Inability or high difficulty in walking is used as an exclusion criterion, how was this determined?

- To describe in more detail, in the case of the inclusion criterion, participate in at least 75% of the training program, which excluded 2 people. It is also stated that the untrained group (UG) consisted of 11 people who did not attend or complete the minimum period of the training programme (i.e. 75%). Why are there 11 of them and not 11 + 2?

How do you then compare the trained group (TG) with the untrained group (UG) if the amount of physical activity performed does not match?

- In anthropometric studies, it should be noted by what methodology BMI and assessment criteria were determined.

- For mathematical statistics, indicate whether the sample followed a normal distribution law.

- The formatting of Table 1 needs to be changed: columns 1 and 2 do not correspond to columns 3 and 4.

- Attention should be paid to the presentation tab. 2 of “rt handgrip” and “lt handgrip” data where the standard deviation is almost 50% of the mean and the presentation of data that does not follow the normal distribution law should be avoided.

- In Fig. 3, add information on the labelling of the two curves.

- In Fig. 5, add information on where Figs. 5A, 5B, etc. are located.

- Provide a description of the conditions for conducting classes.

Author Response

#Reviewer 2

  • This study is aimed at addressing the important issue of improving the health of the elderly as a guarantee of a long healthy life. This study is relevant and up-to-date.

The authors propose a combination of physical exercises and modern telecommunication technologies during classes. Thus, combining classes in a group with a trainer and independent classes at home, which significantly expands the arsenal of influence.

The authors are guided by a wide range of scientific research in recent years. They use modern research methods to assess the health benefits of the proposed training programme.

The structure of the pedagogical experiment developed by the authors meets the requirements for testing this hypothesis.

The conclusions reflect the research findings.

In addition to the positive aspects of the presented article, there are provisions that require additional discussion:

Dear Reviewer, thank you for your evaluation and the suggestions for our study. The comments were helpful to improve the quality of the manuscript.

  • In the introduction, outline the range of unresolved issues that underpin the purpose and objectives of the article.

Dear Reviewer, thank you for this suggestion. This section has been improved accordingly.

  • Inability or high difficulty in walking is used as an exclusion criterion, how was this determined?

Thank you for raising this comment. The inability and high difficulty in walking has been established with the score of the Tinetti Scale. This concept has been clarified in the manuscript.

  • To describe in more detail, in the case of the inclusion criterion, participate in at least 75% of the training program, which excluded 2 people. It is also stated that the untrained group (UG) consisted of 11 people who did not attend or complete the minimum period of the training programme (i.e. 75%). Why are there 11 of them and not 11 + 2?

Thank you for this comment. We apologize for the lack of clarity. The UG is formed by people who have not completed the minimum training period. Only 11 people were included in this group because 2 people dropped-out the study.

  • How do you then compare the trained group (TG) with the untrained group (UG) if the amount of physical activity performed does not match?

Thank you for this comment. The purpose of this study was not to compare the two groups, but to evaluate the effectiveness of a training protocol on the training group. We found an error in the manuscript and made the changes to develop the concept more clearly. The analysis was conducted exclusively within groups.

  • In anthropometric studies, it should be noted by what methodology BMI and assessment criteria were determined.

Thank you for this comment. BMI was calculated indirectly by equation; more details can be found in the “Measurements” section.

  • For mathematical statistics, indicate whether the sample followed a normal distribution law.

We agree with the reviewer. This concept has been explained in the Table 2.

  • The formatting of Table 1 needs to be changed: columns 1 and 2 do not correspond to columns 3 and 4.

Thank you for this suggestion. Table 1 has been formatted.

  • Attention should be paid to the presentation tab. 2 of “rt handgrip” and “lt handgrip” data where the standard deviation is almost 50% of the mean and the presentation of data that does not follow the normal distribution law should be avoided.

Dear Reviewer, thank you for this acute suggestion. We reviewed Table 2.

  • In Fig. 3, add information on the labelling of the two curves.

We apologize for the lack of clarity. The figure 3 has been changed to make it clearer.

  • In Fig. 5, add information on where Figs. 5A, 5B, etc. are located.

Thank you for this comment. We have inserted the relative information in the manuscript.

  • Provide a description of the conditions for conducting classes.

Thank you for this comment. During the supervised activities the participants were divided into two groups and alternated in the execution of the activities. During the telecoaching activities each patient performed the activities in their own home spaces. More details have been included in the manuscript.

Reviewer 3 Report

Comments and Suggestions for Authors

1. What is the main question addressed by the research?

The research intended to investigate the effectiveness of a sustainable training program combining supervised outdoor exercise with telecoaching on physical performance in elderly people. This was fully addressed showing the gaps in the current literature, for example, linking green exercise, psychophysiological health and telecoaching clearly demonstrated in the introduction.

2. Do you consider the topic original or relevant to the field? Does it address a specific gap in the field?

This interdisciplinary and multiapproach topic is original, very related in the post pandemic era. The authors addressed the gaps.

Literature available give more detail on the need of awareness of physiology of training, physical activity/exercise through use of technology. Little has been done to combine the three aspects addressed by the authors. Importantly is finding out a lasting solution ie., - effectiveness and sustainability of training program combining supervised outdoor exercise with telecoaching on physical performance in elderly people.

Authors need to re-work on their materials and methods section. Subheading measurements is not very clear of measurements to be conducted unless if the tests are itemised eg., (a), (b)…… or (i), (ii)……

‘’The measurements were carried out by sports …………’’ This sentence may fit well under study design.

3. What does it add to the subject area compared with other published material?

This does add literature to the available literature. Testing the protocol’s sustainability is important as this addresses a global issue on healthy aging.

4. Are the conclusions consistent with the evidence and arguments presented and do they address the main question posed?

Conclusions address discussed arguments and the objective of the paper.

5. Are the references appropriate?

All references are appropriate. However, authors need re-check the journal referencing as some references are written author + et al., while some are listed mote than 3 authors.  

6. Please include any additional comments regarding tables.

Generally, the tables/ figures are well designed.

Author Response

#Reviewer 3

  • What is the main question addressed by the research?

The research intended to investigate the effectiveness of a sustainable training program combining supervised outdoor exercise with telecoaching on physical performance in elderly people. This was fully addressed showing the gaps in the current literature, for example, linking green exercise, psychophysiological health and telecoaching clearly demonstrated in the introduction.

Dear Reviewer, thank you for the evaluation of our manuscript.

  • Do you consider the topic original or relevant to the field? Does it address a specific gap in the field?

This interdisciplinary and multiapproach topic is original, very related in the post pandemic era. The authors addressed the gaps.

Literature available give more detail on the need of awareness of physiology of training, physical activity/exercise through use of technology. Little has been done to combine the three aspects addressed by the authors. Importantly is finding out a lasting solution ie., - effectiveness and sustainability of training program combining supervised outdoor exercise with telecoaching on physical performance in elderly people.

Dear Reviewer, we appreciated your judgment.

  • Authors need to re-work on their materials and methods section. Subheading measurements is not very clear of measurements to be conducted unless if the tests are itemised eg., (a), (b)…… or (i), (ii)……

Thank you for this comment. The section relative to measurements was as suggested.

  • ‘’The measurements were carried out by sports …………’’ This sentence may fit well under study design.

Thank you for this suggestion. This modify has been made in the new version of the manuscript.

  • What does it add to the subject area compared with other published material?

This does add literature to the available literature. Testing the protocol’s sustainability is important as this addresses a global issue on healthy aging.

Dear Reviewer, thank you.

  • Are the conclusions consistent with the evidence and arguments presented and do they address the main question posed?

Conclusions address discussed arguments and the objective of the paper.

We appreciated, thank you.

  • All references are appropriate. However, authors need re-check the journal referencing as some references are written author + et al., while some are listed more than 3 authors.

Thank you for raising this point. The references were revised according to the journal referencing.

  • Please include any additional comments regarding tables.

Generally, the tables/ figures are well designed.

Thank you for the evaluation.

Reviewer 4 Report

Comments and Suggestions for Authors

Many thanks to the reviewers for considering me for this review and to the authors for the time they have taken to prepare this work.
The following is a series of contributions to be able to contribute to the work with the aim of improving its quality. It is an interesting work that combines supervised outdoor exercise with telecoaching on physical performance in elderly people.
Regarding the abstract, it is necessary to eliminate the word "background". It is also recommended to reduce the acronyms and data in the abstract in order to facilitate reading. The measurement instruments should appear in the abstract.
As for the introduction, there is no in-depth analysis of the variables to be analyzed later. Likewise, it is not in line with the title of the article. This introduction should go into greater depth and clarify very clearly and directly the variables involved in the study and justify their importance.

Regarding the design of the study, it should be called scientifically: is it a quasi-experimental or experimental study? Regarding the sample, it is recommended to provide in a table the data in Figure 1 regarding the trained and untrained group.
Regarding the measurement instruments, it is recommended to relate them to the variables. Likewise, the data referring to the program should be presented in a better, clearer way, and in any case in the procedure section and not in the measurement section.
Regarding the statistical analysis, the authors should provide the values of the shapiro wilk test in such a way as to demonstrate that the parametric tests used are correct. If this is not the case, all the results are questioned because it is not known whether they have used the optimal tests for their objectives.

Regarding table 1. Are the sample description data repeated? It seems to be yes. Data should not be repeated.
This is the most deficient section of the paper since the results are not commented or highlighted and, as indicated above, it is not known whether the correct tests have been used according to the sample distribution. This invalidates the following discussion section.
As for the conclusion half of the section is about recommendations. It should be redone once the results are correct.

Author Response

#Reviewer 4

  • Many thanks to the reviewers for considering me for this review and to the authors for the time they have taken to prepare this work.

The following is a series of contributions to be able to contribute to the work with the aim of improving its quality. It is an interesting work that combines supervised outdoor exercise with telecoaching on physical performance in elderly people.

Dear Reviewer, thank you for the evaluation of our manuscript. The comments were very helpful to improve the quality of the manuscript.

  • Regarding the abstract, it is necessary to eliminate the word "background". It is also recommended to reduce the acronyms and data in the abstract in order to facilitate reading. The measurement instruments should appear in the abstract.

Thank you for this suggestion. The entire abstract section has been revised accordingly.

  • As for the introduction, there is no in-depth analysis of the variables to be analyzed later. Likewise, it is not in line with the title of the article. This introduction should go into greater depth and clarify very clearly and directly the variables involved in the study and justify their importance.

Thank you for this acute comment. The introduction section has been revised according to these suggestions.

  • Regarding the design of the study, it should be called scientifically: is it a quasi-experimental or experimental study?

Dear Reviewer, thank you for this insightful comment. In this study we did not randomly assigned participants to the experimental or control group. Hence, our study has a quasi-experimental design.

  • Regarding the sample, it is recommended to provide in a table the data in Figure 1 regarding the trained and untrained group.

Thank you for this suggestion. In accordance with this comment, Figure 1 has been changed.

  • Regarding the measurement instruments, it is recommended to relate them to the variables. Likewise, the data referring to the program should be presented in a better, clearer way, and in any case in the procedure section and not in the measurement section.

Thank you for this comment. The measurements section has been revised and improve according to this suggestion.

  • Regarding the statistical analysis, the authors should provide the values of the shapiro wilk test in such a way as to demonstrate that the parametric tests used are correct. If this is not the case, all the results are questioned because it is not known whether they have used the optimal tests for their objectives.

We agree with the reviewer, and we thank the reviewer for this comment. Indeed, the Student T-test or Wilcoxon T-test were used in relation to the results of the normality test. More details were included in the section "statistical analysis" and in Table 2.

  • Regarding table 1. Are the sample description data repeated? It seems to be yes. Data should not be repeated.

Thank you for this suggestion. Table 1 has been revised based on the comment.

  • This is the most deficient section of the paper since the results are not commented or highlighted and, as indicated above, it is not known whether the correct tests have been used according to the sample distribution. This invalidates the following discussion section.

Dear Reviewer, thank you for raising this point. The tables and figures of the result section has been revised in order to clarify the test used and final results.

  • As for the conclusion half of the section is about recommendations. It should be redone once the results are correct.

We apologize for the inconvenience, and we agree with the reviewer. The conclusion section has been revised and improved.

Round 2

Reviewer 1 Report

Comments and Suggestions for Authors

The authors adressed all required changes

Author Response

Dear Reviewer, thank you for your evaluation.

Reviewer 4 Report

Comments and Suggestions for Authors

Thank you for the modifications.

More substantial modifications were required. Those made are very superficial.

Round 3

Reviewer 4 Report

Comments and Suggestions for Authors

My comments have already been made. It is up to the editor of the journal to make the decision.

Author Response

Dear Academic Editor,

thank you for this additional note which aims to further improve our work.

We are pleased that we have addressed the comments of 3 of the 4 reviewers.

Regarding the comments of reviewer 4, we have revised every single point in the previous round of revision. In addition to the changes already made, we have now argued a further systematic review on the topic to better cover the background, modified some parts of the discussion, and inserted a further limitation regarding the study design (the fact that we did not randomize participants between the two groups and this therefore makes it a quasi-experimental study).